# Analysis of Tool Wear and Hole Delamination for Large-Diameter Drilling of CFRP Aircraft Fuselage Components: Identifying Performance Improvement Drivers and Optimization Opportunities

Juan Fernández-Pérez [ID], Carlos Domínguez-Monferrer *, María Henar Miguélez and José Luis Cantero [ID]

Department of Mechanical Engineering, Universidad Carlos III de Madrid, Avda. de la Universidad 30, Leganés, 28911 Madrid, Spain; juanfp4@gmail.com (J.F.-P.); mhmiguel@ing.uc3m.es (M.H.M.); jcantero@ing.uc3m.es (J.L.C.)
* Correspondence: cardomin@ing.uc3m.es; Tel.: +34-916248873

**Abstract:** This study provides a comprehensive analysis of the one-shot drilling (CFRP) strategy for machining CFRP materials in the assembly of aircraft components, focusing on key factors such as tool wear, hole delamination, and the evolution of machining forces. The research uses adapted parts of the tail-cone structure of a commercial aircraft as workpieces and employs large-diameter cutting tools to perform drilling operations, with results that can be readily applied to the industry. The study selects cutting conditions by analyzing the effect of cutting parameters on tool life in drilled holes and accumulated cutting time, with the end-of-life criterion based on the extension of the wear suffered by the main cutting edge of the first step. The results show that all tested cutting conditions achieve a similar value of tool life expressed in terms of holes drilled, with differences smaller than 7%. However, one of the cutting conditions analyzed completes the same number of holes within 40% less time. Therefore, considering productivity criteria, it will be interesting to evaluate the use of high values for the cutting parameters. Overall, this research provides valuable insights for improving the efficiency and effectiveness of CFRP machining in aircraft manufacturing.

**Keywords:** one-shot drilling (OSD); CFRP drilling optimization; hole quality; tool wear; aircraft assembly process

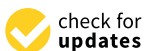



## 1. Introduction

Market needs and customer expectations are significantly transforming how today's aircraft are designed and manufactured. New services will be driven by sustainability and enabled by innovation, driving efficiency, and supporting net-zero $CO_2$ emissions targets. According to the aviation manufacturer Airbus, by 2041, new-generation passenger aircraft will represent 95% of the operated fleet. Considering that the most modern aircraft, such as the Airbus A350 or the Boeing 787, are manufactured with a structural weight of composite materials of approximately 50%, this value is expected to increase in the future, especially the use of carbon-fiber-reinforced polymers (CFRPs). Their high strength-to-weight ratio, corrosion resistance, and low thermal expansion coefficient make them attractive in aircraft structures, including wings, fuselages, and tail sections [1–3].

Typically, they are made by layering sheets or fabric of carbon-fiber material and embedding them in a polymer matrix, such as an epoxy resin. The layers are then cured under heat and pressure to create a strong, lightweight composite material. Compared to traditional materials such as aluminum or steel, a CFRP is much lighter and has a lower carbon footprint, making it more attractive to manufacturers and operators who are looking to reduce their environmental impact, improving the fuel efficiency of aircraft [4–7].

Despite their potential benefits, using CFRPs in aircraft structural components is challenging. A CFRP material is heterogeneous, consisting of two phases, matrix and fibers, which have quite different properties that can affect the drilling process and the quality of the workpiece, causing machining-induced damage. This damage cannegatively impact

mechanical joints' reliability and components' structural integrity [8]. Delamination is the most significant concern among the main defects that can arise among, including uncut fibers, hole surface damage, and thermal degradation [9,10]. This phenomenon is caused by the low interlaminate strength of the composite structure and the excessive thrust force produced during machining [11]. In addition, carbon fibers' hardness and abrasiveness lead to severe wear on cutting tools, increasing cutting forces, worsening hole quality and dimensional stability, and, in general, limiting the useful life of the cutting tool [12]. These challenges require careful consideration and management to ensure that the full potential of CFRPs in aircraft manufacturing can be realized.

During the assembly of aeronautical components, drilling operations play a critical role. The tools' lifespan in these operations is determined by various factors such as diameter, roughness, and delamination. Typically, the end of a tool's life is defined by the point at which the last drilled hole no longer meets the required specifications. This study specifically focuses on evaluating the quality of drilled holes in relation to delamination, which is the most significant issue.

On the other hand, in the context of the aircraft industry, drilling processes are one of the most important operations in component assembly. Boeing's 747 has more than 3 million mechanical joints to assemble its 6 million components, while in the case of the Airbus A380, the wings alone have 32,000 parts and 750,000 rivets. The strict quality and safety standards, the high value of the parts, and the additional complexity of drilling stacks make the cost-per-hole highly variable. Hence, scientific studies in this field are particularly interesting for higher production speed, lower tool wear, top-quality standards, and a reduction in the use of consumables to reduce the associated costs, time, and environmental footprint [13,14].

Thus, the main challenge facing the aeronautical sector in the aircraft assembly stage for the near future is to optimize the CFRP drilling processes. Dozens of articles on the optimization of composites drilling operations can be found in the literature. It is usually evaluating the tool wear and hole delamination under different cutting conditions, comparing traditional and special drill geometries with tool diameters varying between 4 and 8 mm in most cases, as they are widely used in the aerospace industry [15–33]. As for the materials used in the drills, cemented carbides (*Tungsten Carbide*, WC) provide one of the best economic performances because of their high strength, adequate abrasion resistance, ability to withstand high temperatures, and their ability to provide a good balance between hardness and toughness [34,35]. WC substrates are generally diamond-coated to improve their durability by providing additional hardness and wear resistance [36,37] and modifying the tool wear mechanism [38]. While uncoated carbides mainly suffer cutting-edge rounding due to abrasion [39], which produces a constant cutting force increase with tool wear [40], the addition of the coating causes a combination of chipping and delamination of the diamond layer and a further abrasion of the substrate [41,42].

In this context, bringing novelty to this field of study is challenging, given that hundreds of drilling configurations have already been tested. However, a similar investigation-gap pattern has been observed in all of them.

The studies focus on drilling a CFRP using tools recommended by different manufacturers. In some cases, special geometries are proposed that still need to be implemented in the aeronautical sector, making it unclear whether they would be valid in real working conditions in terms of durability or quality of the component. Given the high levels of safety and component quality required in this industry, industry-specific tools are typically needed to optimize CFRP drilling under real conditions. In most cases, cutting tools should come directly from a real production system to ensure suitability. For instance, in the aircraft industry, countersinking is typically included during the drilling operation to avoid interference with the aerodynamic shape of the rivet head. Therefore, studies oriented toward this sector should consider cutting tools with this configuration to ensure the results are relevant to real-world conditions.

As for the workpieces, their thickness, the number of sheets, and their orientations vary greatly depending on each investigation. Therefore, the wear analyzed in the tools as

well as the damage induced in the component, is not easily extrapolated to the industry if we do not work with conditions similar to those of a production system. The same applies to the various cutting parameters that are analyzed.

On the other hand, to establish limited damage in the drilled component, the quality standards followed in the sector to which the research is oriented must be followed. For example, in the aeronautical field, the manufacturer's Qualified Test Method List and Nadcap commodities must be followed. It has been observed that in many studies, the basis used to establish these criteria is not clear.

Finally, the influence of the cutting parameters on tool life should also be analyzed, both in minutes and in the number of holes per cutting edge, the latter being more directly applicable to increasing machining productivity. In most cases, research is focused on determining the best cutting parameters or geometries that minimize tool wear and component damage, regardless of the time taken to drill the holes, with the consequent impact on aircraft production.

Nevertheless, studies comparable to the one presented in this manuscript can also be found in the literature.

Rahme et al. [43] studied the delamination at the exit of the hole when drilling a thick composite laminate using a gun drill of a diameter of 16 mm. The study aimed to determine a critical feed rate per tooth to avoid delamination. For this purpose, a drilling thrust force model was considered to calculate the drilling forces of each zone concerning the feed rate per tooth. Compared to the present study, the geometry and dimensions of the drill bit used, as well as the field of application of the drill bit, since it was specified that it was valid for all types of materials, were notably different. Moreover, the reason for the choice of the workpiece configuration was not given, and the impact of the optimal cutting parameters with the hole-making time was not considered.

Priarone et al. [33] considered the productivity of the process and analyzed the workpiece delamination, thrust force, torque, and chip morphology under different cutting conditions with 6 mm core drills. In this sense, although the dimensions and geometries of the drill differed, the variables monitored, and the considerations taken were similar to those of the present study. However, the cutting parameter range evaluated was chosen according to the typical values applied by various researchers when using core drills and did not consider whether the values were applicable to the industry. Likewise, the reason for the choice of the workpiece configuration was not given.

On the other hand, Khasaba et al. [44] investigated the effect of machining parameters on the machinability parameters in drilling thin woven glass-fiber-reinforced epoxy laminates using a standard 6.5 mm coated twist drill. The thrust force and torque, temperature, delamination (peel-up at entry and push-out at exit), bearing strength, and surface roughness were assessed. Although that study was the most comprehensive of those mentioned above, the drilling of woven glass-fiber-reinforced epoxy composites (GFRE) is not as widespread in the aerospace industry as CFRPs. In addition, the difference in mechanical properties of GFRE and CFRPs makes their machinability and machining-induced damage different. A similar study was performed by the same authors in [45,46] and, in that context, a hybrid approach was the study conducted by Rahme et al. [47], where the drilling of a carbon/epoxy composite laminate with one glass-woven fabric ply at the exit of the hole was studied.

These gaps constituted the main motivation for the present work. The aim was to identify the performance improvement drivers and optimization opportunities of the one-shot strategy during the drilling of CFRPs in the assembly of aircraft components, with the main contributions being as follows:

(1) The cutting tools studied are used in the tail-cone assembly phase of a commercial aircraft. These are double-pitch twist drills with a diameter of 9.5 mm. During the literature review, it was found that the number of investigations where drills between 8 and 12 mm had been used was very limited compared to the 4–8 mm range [48–50], finding similar investigation-gap patterns as explained above.

(2)   The workpieces were adaptations of components from the upper quadrant of the tail-cone structure of a commercial aircraft, while the cutting parameters analyzed were based on the ones used in aircraft assembly processes and the manufacturer's recommendations. Therefore, the study results can potentially be extrapolated to the industry.

(3)   An analysis of the effect of cutting parameters on tool life, both in drilled holes and accumulated cutting time, was carried out to select those cutting conditions that had the least impact on a real production system. It was necessary to find a compromise solution that allowed a large number of holes to be drilled quickly without compromising the component's quality. This was considered in optimizing the drilling process through a combined, comprehensive, and in-depth analysis of the main elements involved in the machining process: tool wear, hole delamination, and the evolution of the machining forces components.

## 2. Materials and Methods

### 2.1. Workpiece Material

The test specimens were adapted from a part of the assembly process of an aircraft fuselage component. Hence, the regulations and requirements of the aviation sector were fulfilled. The CFRP consisted of several unidirectional carbon fiber sheets pre-impregnated with epoxy resin with a resin content of 34%. In all cases, the first layer, corresponding to the skin of the aircraft, had a coating of expanded copper, which acted as protection against lightning strikes. An additional layer of fiberglass fabric was added to the bottom of this layer to mitigate the appearance of machining-induced damage. Both of them were pre-impregnated with an epoxy resin.

Table 1 shows the manufacturing conditions of the specimen material and the mechanical properties of the carbon-fiber sheets. Before drilling, the specimens were cut in a waterjet machine into an A4 format according to the international standard ISO 216 (210 × 297 mm). The samples had a thickness of 14.5 mm and consisted of 85 sheets with the following orientations: 23.1% at 0 degrees, 38.5% at 90 degrees, 19.2% at 45 degrees, and 19.2% at 135 degrees.

**Table 1.** Mechanical properties of the carbon-fiber sheets and manufacturing conditions of the specimens.

| Processing Conditions | | Layer Mechanical Properties | |
|---|---|---|---|
| Cure temp. (°C) | 180 | Tensile strength 0° (MPa) | 2750 |
| Cure pressure (MPa) | 0.1–1 | Tensile strength 90° (MPa) | 30 |
| Vacuum (MPa) | 0.01–0.1 | Compression strength 0° (MPa) | 1200 |
| | | Compression strength 90° (MPa) | 170 |
| | | Interlayer shear strength (MPa) | 70 |
| | | Fiber density (g/cm3) | 1.78 |

Throughout the fuselage of the aircraft, different CFRP configurations are used depending on the structure: frames, ribs, stringers, or skin. In this case, this specification was selected because it was the one in which the analyzed tool was used in the real production system.

### 2.2. Tool Geometry and Machining Conditions

The cutting tools were two-edged twist drills with a 40-degree right-hand helix and highly positive rake angles. The tip geometry had a split-point sharpening with a tip angle of 90 degrees. The geometry had two steps with maximum diameters of 8.2 mm and 9.54 mm, respectively. This is common in the industry, and several studies have shown that splitting the main cutting edge performs better on large-diameter drills [51–53]. The distance of the second step with respect to the tip of the tool was 7.1 mm. These were tools manufactured by Ham Präzision with a special design for the drilling process of aircraft structures (see Figure 1).

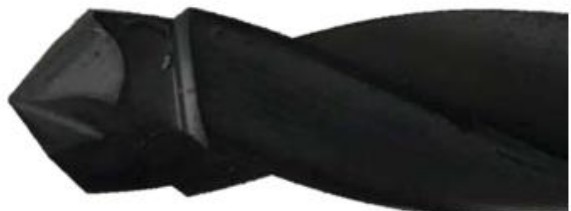

**Figure 1.** Diamond-coated carbide drill bit.

The tool substrate was EMT100, based on a micrograin-cemented carbide with an average grain size of 0.8 μm. It comprised 93% tungsten carbide, 6% cobalt, and 1% other materials.

To analyze the influence of the parameters on the efficiency of the composite drilling process, the cutting speed and feed were varied independently by a factor of 20%. These, presented in Table 2, were chosen based on the ones used in aircraft assembly processes and the tool manufacturer's recommendations. All tests were carried out under dry conditions, similar to the industrial operation. A total of 285 holes were drilled in each test. The tool wear, thrust force, and torque were checked at 70, 140, 210, and 285 holes, as well as on the first hole, for the cutting conditions tried.

**Table 2.** Cutting parameters tested with 9.5 mm diameter tools in CFRP drilling.

| Test | Rotation (rev/min) | Cutting Speed (VC) (m/min) | Feed Speed (Vf) (mm/min) | Feed (f) (mm/rev) |
|------|--------------------|----------------------------|--------------------------|-------------------|
| 1 | 1668 | 50 | 167 | 0.1 |
| 2 | 2002 | 60 | 200 | 0.1 |
| 3 | 2336 | 70 | 234 | 0.1 |
| 4 | 1668 | 50 | 200 | 0.12 |
| 5 | 2002 | 60 | 240 | 0.12 |
| 6 | 2336 | 70 | 280 | 0.12 |

The tool life was quantified as a function of cumulative cutting time because it is the most commonly deployed in wear-test analyses and in determining analytical expressions for tool life.

*2.3. Hole Delamination*

The holes were inspected with a C-Scan and no internal damage was observed. Only delamination and uncut fibers were observed at the exit of the hole, so the delamination factor was used to quantify it.

It was defined as the ratio of the damaged diameter ($D_{damage}$) over the nominal diameter of the hole ($D_{nominal}$), as stated in Equation (1).

$$D_f = \frac{D_{damage}}{D_{nominal}} \tag{1}$$

To determine the delamination factor, three measurements were taken at the entrance of the hole, and three at the exit, and the average between them was taken as the final value. It should be noted that the hole diameter measurement was only used to determine the delamination factor. However, no specific quality control of the hole diameter was performed, so the industry standard of 4 measurements per hole was not applied. The measurements were performed according to Airbus's norms and standards.

Due to the relevance of delamination to the functionality and integrity of a structural element composed of composite materials, it can be examined in further detail. Samborski [54] suggested an analytical technique to explain the beginning, growth, and direction of the delamination front. Feito et al. [55] studied and examined the application

of the delamination factor to evaluate the effectiveness of a drilling operation as one of the more experimental ways.

### 2.4. Experimental Resources

The tests were carried out on a 3-axis KONDIA B500 machining center with a Heidenhain numerical control. A hydraulically expanding chuck DIN 69871 with a clamping diameter of 20 mm was used for holding the cutting tools. In addition, slotted bushings for hydraulic expansion cones were used to adapt them to the different tool shank diameters.

To monitor the process and analyze the cutting forces and torques, a Kistler piezoelectric rotary dynamometer, model 9123C, was implemented with an acquisition frequency of 2000 Hz. Hence, an amplifier (Kistler 5070A) was also needed to send a signal in the form of an electrical voltage ($\pm 10$ V), which was captured by a signal acquisition card (Keithley Kusb 3100 m) and sent to a computer, via Dyno Wire software, for processing.

Regarding tool wear monitoring, an optical microscope model OPTIKA SZR was used with an optical camera and Leica software to process the images, together with a Philips XL-30 scanning electron microscope with an EDSDX4i system.

Additionally, the performance of the tests required the design, fabrication, and assembly of custom tooling to encapsulate the dust produced during the drilling process on both sides of the specimen and to simulate machining conditions encountered in the industry (positioning of specimens and concentric suction around the spindle and tool). Because of the need to vacuum the dust produced during machining from both sides of the specimen, it was decided to manufacture the tooling in two parts with a vacuum hose connected to each one. The model of the industrial cleaner used was Nilfisk S2B with a cyclonic filter, antistatic filters, and a microfilter (HEPA class H) to retain the smallest particles.

## 3. Results

The influence of cutting parameters on the drilling performance of drills in terms of tool wear and evolution of cutting forces, as well as the tool life obtained both in terms of holes and cumulative cutting, was analyzed to select those cutting conditions that had the least impact on a real production system.

### 3.1. Tool Wear Evolution

As indicated in Section 2.2, 9.5 mm diameter holes were machined using step drills. Figure 2 shows in detail the wear at the corner of the cutting edge of the first and second steps. This mechanism was very similar to that undergone by the smaller-diameter drills investigated in other studies [56,57]. However, particular features were observed.

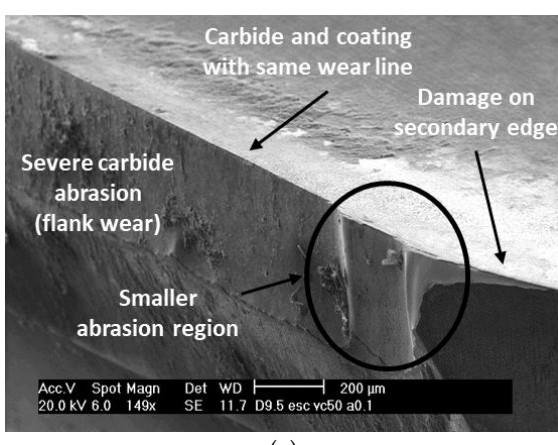
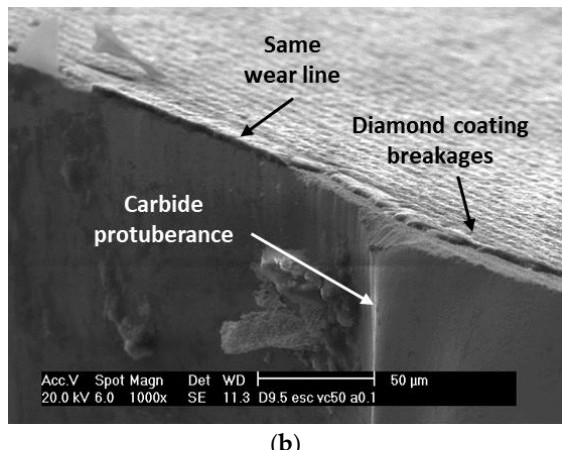

(**a**)        (**b**)

**Figure 2.** Wear suffered on the cutting edge of the first step when drilling with $V_C$ = 50 m/min and f = 0.10 mm/rev after 285 holes. (**a**) Corner of the cutting edge; (**b**) detail of substrate area with a lower wear level.

The main wear mechanism was located at the flank, combined with small breakages in the coating. The losses of the diamond coating were produced due to the interaction with the highly abrasive carbon fiber particles and fluctuating cutting forces. Combining both effects led to a stress concentration, promoting cutting-edge weakening and discontinuous breaks, especially at the corner of the main cutting edge. In addition, as shown in Figures 2 and 3, there was no rounding of edges in the tool.

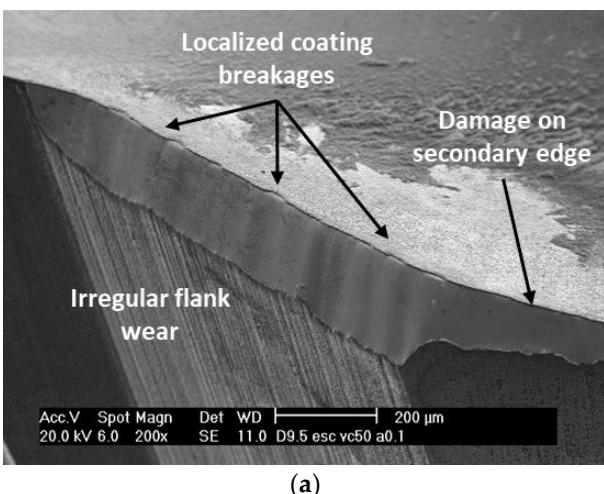
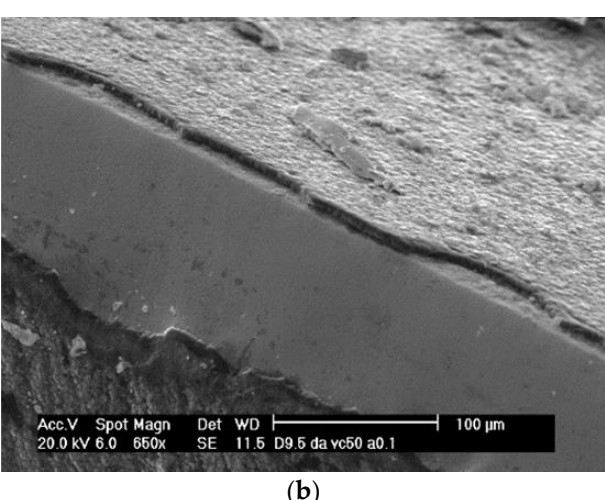

(**a**)　　　　　　　　　　　　　　　　　　　　　　　　　　　　(**b**)

**Figure 3.** Wear suffered on the cutting edge of the second step when drilling with $V_C$ = 50 m/min and f = 0.10 mm/rev after 285 holes. (**a**) Corner of the cutting-edge; (**b**) detail of the diamond coating breakage.

The occurrence of the flank was diminished at the region near the tool tip due to the reduced cutting speed, close to the axis of rotation. In addition, as wear progressed, the sharpness of the cutting edge was reduced due to rounding.

The flank wear on the main cutting edge of the first step was produced due to the breakage of the coating and subsequent abrasion of the substrate of relative uniform width (see Figure 2a). When approaching the corner of the cutting edge, larger coating breaks and a greater wear irregularity were observed due to the higher speed at which the fibers hit the tool. Additionally, Figure 2b shows a singular form due to the fact that, although there was a significant loss of coating, the substrate suffered less abrasive wear than in areas close to the cutting edge, a protrusion of the substrate. This phenomenon occurred in areas close to the corner of the cutting edge where the cutting speed and abrasion of the material was maximum. Therefore, it was reasonable to consider that it was due to a higher hardness of the substrate in that area related to the manufacturing process of the tool. All in all, to achieve proper adhesion of the diamond coating, it was necessary to pretreat the substrate to reduce the cobalt content. However, this resulted in a reduction of the hard metal strength in the uppermost layers [58,59]. Some particularity in the substrate-coating interface along this process could alter the material properties at the corner of the cutting edge, explaining the lower abrasion in that area.

Furthermore, higher brittle breaks of the diamond coating occurred at the cutting edge of the second step on the rake surface, creating an irregular surface corresponding to flank wear of variable width (see Figure 3a,b). The maximum extension of the damage suffered in the first step was of the order of 400–450 μm, whereas in the second step, the observed flank did not exceed 200 μm. This was due to the lower stiffness of the material on which the cutting edge of the second step worked due to the previous drilling performed by the cutting edge of the first step.

In addition, damage in the second edge of both steps of the tool was produced. In that area, wear was observed due to the coating breakage and abrasion of the substrate.



However, the extension was greater in the second step because it worked with a higher cutting speed due to the larger tool diameter.

In the tests carried out with different cutting parameters, the same wear mechanisms, and particularities were obtained, although there were disparities in the evolution of each of them. To evaluate these differences, measurements of the flank extension were performed according to the ISO 8688 and ISO 3685 standards.

In Figure 4, a rapid increase of the flank extension during the first 4–5 min after the initiation of the flank wear (approximately after a cutting time of 2 min) can be observed. At that point, the rate of wear growth slowed down and was maintained until it reached 200 μm. From that point, the wear evolution accelerated significantly.

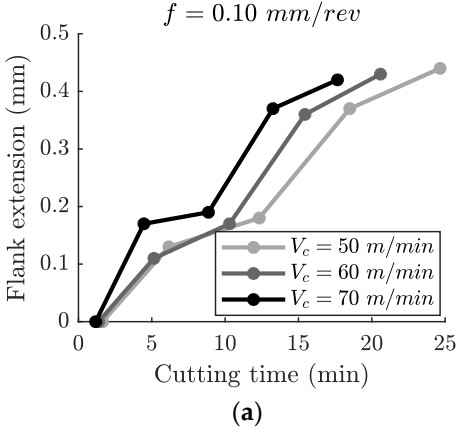 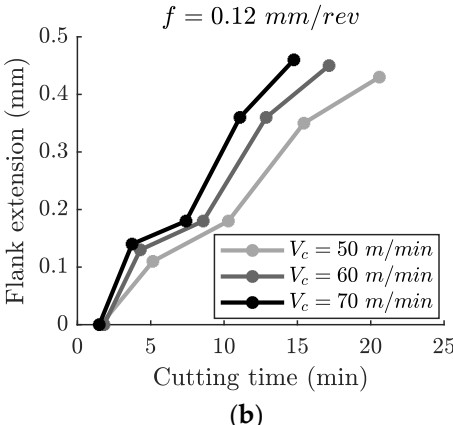

(a)　　　　　　　　　　　　　　　　　　　(b)

**Figure 4.** Influence of cutting parameters on the evolution of the flank extension with cutting time. (**a**) f = 0.10 mm/rev; (**b**) f = 0.12 mm/rev.

On the other hand, the cutting parameters had a significant effect on wear evolution. The higher the cutting speed, the greater the extent of damage relative to the cutting time. This is well known in machining operations, in particular in composite machining, and is related to the more aggressive interaction of the fibers with the rake surface, causing more breaks in the coating and the increased abrasive effect. When the flank wear damage reached 200 μm, a 20% increase in cutting speed resulted in an increase of the flank of between 10% and 20%, depending on the feed and the cutting time.

It was also observed that an increase in the feed increased tool wear because of the larger chip section to be machined. Consequently, in the severe wear region, a 20% feed increase implied a 10% to 20% higher flank.

To sum up, the evolution of the types of wear described was progressive and homogeneous throughout the test, giving rise to a relatively linear evolution of the extent of flank wear with cutting time, as shown in Figure 4. There was an initial stage of relatively rapid wear known as the cutting edge settling and an intermediate zone of linear wear-level growth with some oscillations in the slope due to the fact that only four evaluations were performed per tool.

### 3.2. Thrust Force and Torque Analysis

To evaluate the deterioration of cutting performance due to wear and to analyze the impact of cutting parameters, the thrust force (axial component) and torque were recorded. However, the inherent noise level of signals obtained required a preprocessing technique. According to the literature, there is no single suitable method as it depends on the monitoring procedure and scope. Some examples are wavelet transform, fast Fourier transform, low-pass filter, or power spectrum density. In this work, a first-order low-pass filter was applied with a cut-off frequency of 10 Hz, after several analyses at other frequenciesto remove high-frequency noises while keeping enough information from the source signal.

Figure 5 shows the evolution of the thrust force and torque with respect to the hole depth for a drilling operation with the conditions $V_c$ = 50 m/min and $f$ = 0.10 mm/rev. Due to the presence of the cutting edge of the second step, it can be seen that for a depth of cut ($d_c$) of 7.1 mm, there was an increase in the force and torque, coinciding with the entry of that region into the material. Once all the cutting edges were working on the component, from $d_c$ = 8 mm, there was a progressive decrease in the axial force due to the reduction in its strength since the number of layers at the end of the hole decreased. Another characteristic feature happened at $d_c$ = 14.5 mm, when the drill bit tip exited. It can be seen that the decrease of the thrust force was much more accentuated than that of the torque because the drill bit continued to rub against the borehole wall after this was completed.

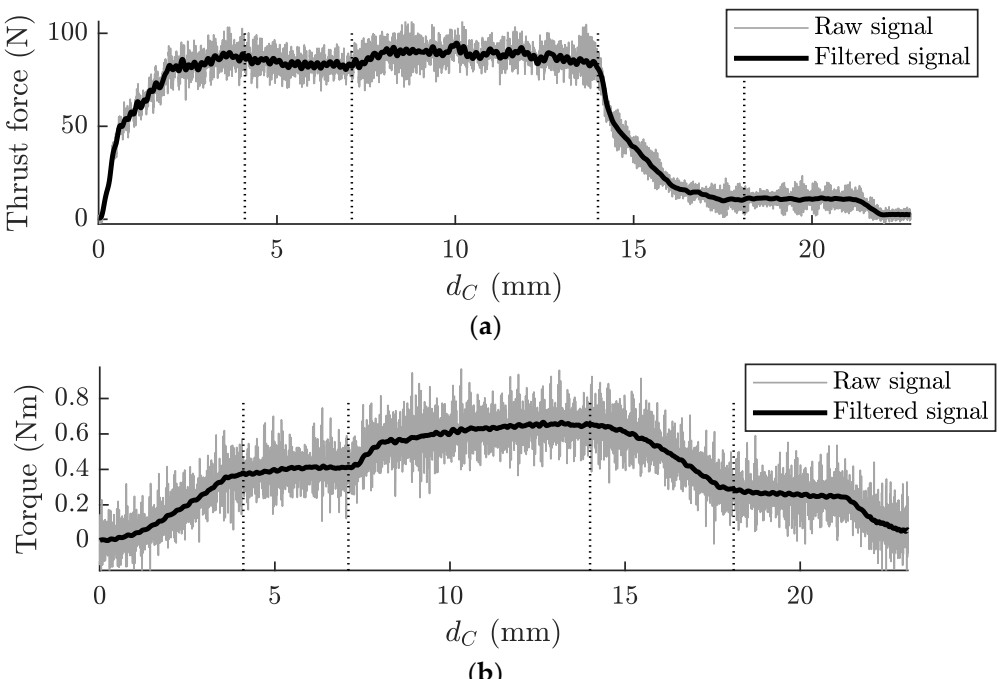

**Figure 5.** Evolution of thrust force (**a**) and torque (**b**) when drilling with $V_C$ = 50 m/min and f = 0.10 mm/rev.

The contribution of the cutting edge of the second step can be seen in Figure 5 from $d_c$ = 18.6 mm, at which point the cutting edge of the first step had completely exited the material. To reduce the effect of the friction of the tool with the walls of the hole, it was necessary to compare the values of the signals in that position with the values at which they stabilized once the drilling process was finished. This contribution was 10% for the axial force and 35% for the torque. Despite the shorter cutting edge's length, the contribution to the torque was significant since that zone produced the highest cutting speeds and generated a higher torque since it was farther away from the rotation axis of the drill bit.

Figure 6 shows the influence of the cutting parameters on the evolution with the cutting time of the thrust force. It can be seen that the force increased linearly with the accumulated cutting time, from the beginning of the tool life and with a constant slope, which depended on the parameters of each test, i.e., with $V_c$ = 50 m/min and $f$ = 0.10 mm/rev, the slope of the thrust force was 31 N/min, while for $V_c$ = 70 m/min and $f$ = 0.10 mm/rev, it increased to 44 N/min (see Figure 6a). On the other hand, in the tests carried out with a feed $f$ = 0.12 mm/rev (see Figure 6b), the slopes obtained were 35 N/min for $V_c$ = 50 m/min, 44 N/min for $V_c$ = 60 m/min, and 50 N/min for $V_c$ = 70 m/min. The influence of increasing the feed from $f$ = 0.10 mm/rev to $f$ = 0.12 mm/rev on the thrust force's growth was greater for higher cutting speeds. The increase in the slope was 13%, 25%, and 14% for cutting speeds of 50 m/min, 60 m/min, and 70 m/min, respectively.

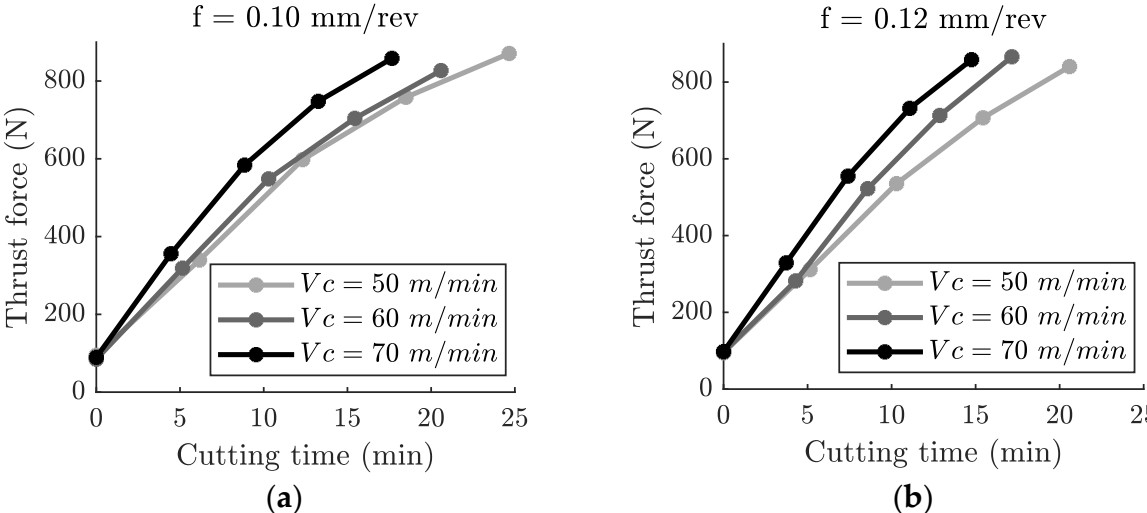

**Figure 6.** Influence of the cutting parameters on the evolution of the thrust force with cutting time from 1 hole to 285 holes. (**a**) f = 0.10 mm/rev; (**b**) f = 0.12 mm/rev.

Figure 7 shows the effect of the cutting parameters on the growth of the torque with cutting time. Its evolution showed two regions with different behaviors. From the beginning of the test, there was a rapid increase in torque until 5 min of cutting time, which corresponded to the initiation of significant flank wear. From this point on, the torque continued to increase linearly, although at a slower rate. This behavior was also observed by other authors [27,38,60].

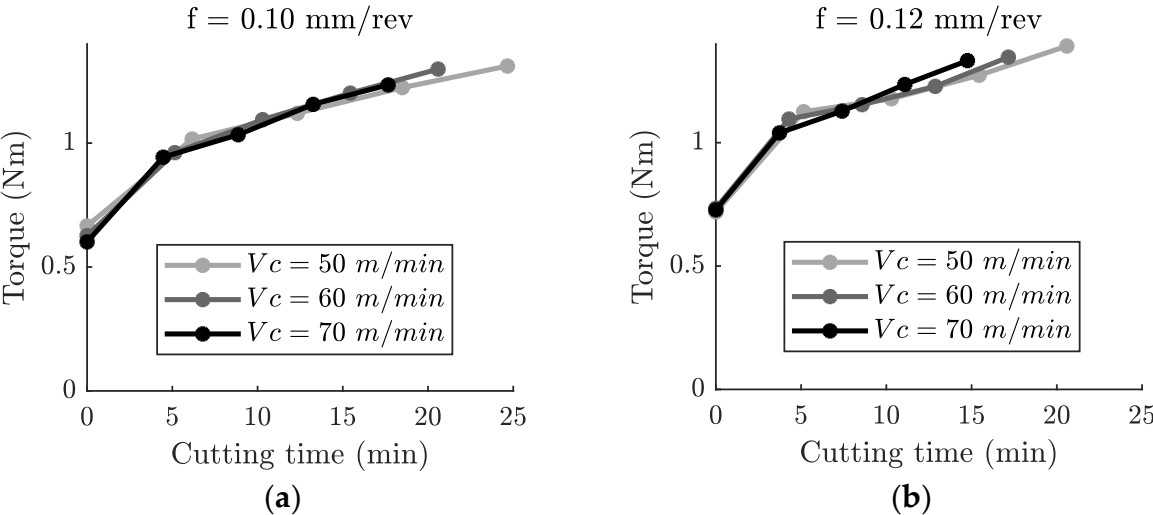

**Figure 7.** Influence of the cutting parameters on the evolution of torque with cutting time from 1 hole to 285 holes. (**a**) f = 0.10 mm/rev; (**b**) f = 0.12 mm/rev.

Figure 8 shows the torque recorded in all tests. In this case, the values corresponding to a feed $f$ = 0.10 mm/rev are identified with light gray and those corresponding to a feed $f$ = 0.12 mm/rev with dark gray. It can be seen that the feed does affect torque from the beginning of the tool's life. The higher the feed, the higher the torque, producing differences between 9% and 12% because of the larger chip cross section. However, the difference was maintained throughout the cutting time tested. This showed that the increase in the feed did not influence the effect of wear on torque, or, in other words, on the slope with cutting time.

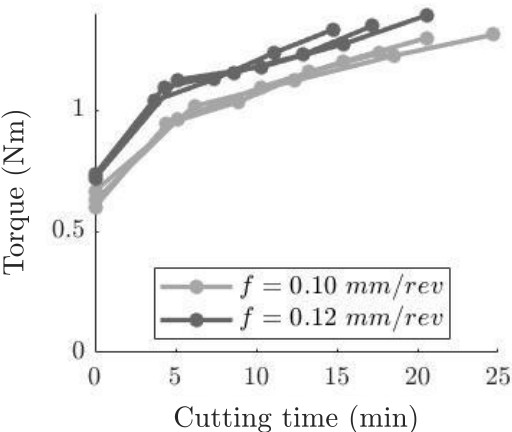

**Figure 8.** Combined plot of the influence of the cutting parameters on the evolution of torque with cutting time from 1 hole to 285 holes.

An important difference between the axial force and torque was their increasing trend. From a new tool to a cutting time of 15 min, they increased by 700% and 100%, respectively. This behavior is due to the fact that flank wear, which is the dominant wear type in these processes, produces a greater increase in the thrust force than in the cutting force, which is directly related to torque. For this reason, monitoring the axial force in these drilling processes could be used to predict the level of tool wear in real time.

### 3.3. Drilling-Induced Damage

Machining-induced damage was affected by tool wear: the main defects detected were delamination and uncut fibers or flaking at the exit of the hole. No significant damage was observed at the entrance of the hole.

Figure 9 shows the evolution of the delamination and uncut fibers at the hole exit. As the cutting time increased, the delamination became more evident. It can be seen that beyond a delamination factor of 1.5, the overall quality of the hole significantly worsened, producing flaking around the perimeter of the hole exit. A severe deterioration of the cutting tool caused this.

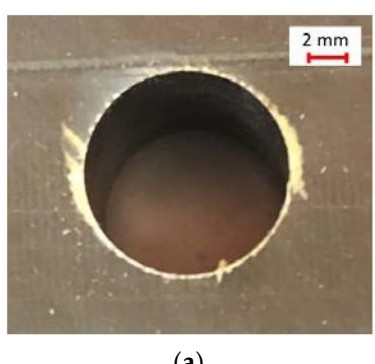

(**a**)

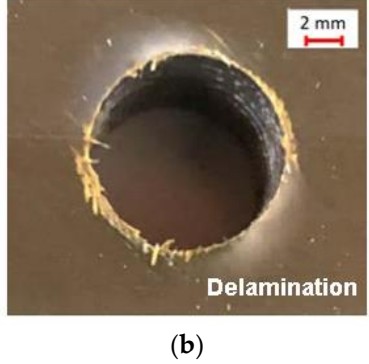

(**b**)

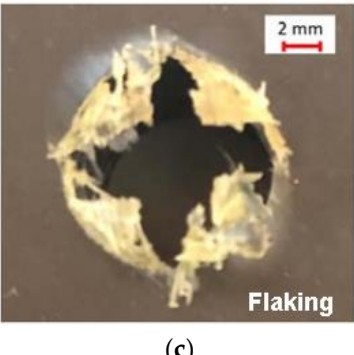

(**c**)

**Figure 9.** Quality evolution at the hole exit when drilling with $V_C$ = 50 m/min and f = 0.10 mm/rev. (**a**) $t_C$ = 5 min and delamination factor of 1.40; (**b**) $t_C$ = 15 min and delamination factor of 1.50; (**c**) $t_C$ = 25 min and delamination factor of 1.55.

Ensuring high levels of safety is crucial, particularly in the aeronautical sector, where conservative tool replacement criteria must be established based on factors such as machining quality and tool wear. It is essential to keep in mind that tool wear not only impacts machining quality but also increases the risk of catastrophic tool breakage. It was observed that the delamination stabilized for moderate cutting times. Therefore, the delamination did not allow the establishment of an operative tool replacement criterion, and the end of

the tests was defined when a maximum value of 0.4 mm in the flank wear extension of the main cutting edge of the first step was reached as this was the dominant type of flank wear extension of the first-step main cutting edge.

Figure 10 shows the evolution of the delamination factor at the hole exit. In this case, a point cloud is shown with each measurement made. Although the evolution of the delamination factor with the cutting time was the same for all of them, it can be seen that there was no clear trend with the cutting parameters.

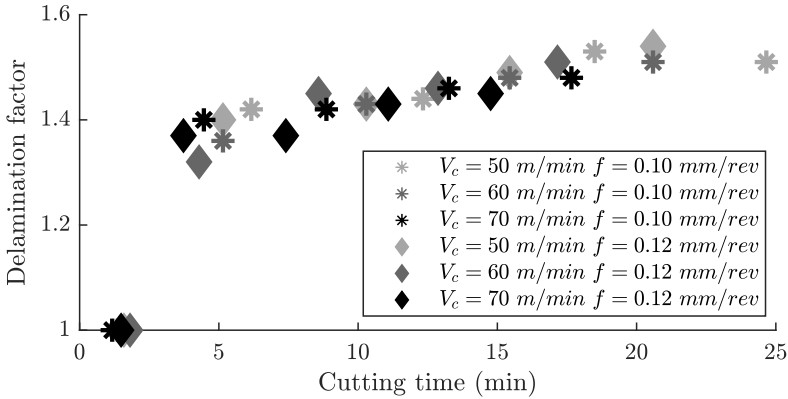

**Figure 10.** Evolution of the delamination factor at the hole exit with cutting time for all cutting conditions tested.

Initially, no defects were produced in the hole until about 2 min of cutting time was reached, when delamination damage started to become noticeable. From this point on, the delamination factor grew rapidly, reaching a value of 1.4 at around 5 min of cutting time. At that point, uncut fibers were produced along the entire perimeter of the hole, which led to a general deterioration in the quality of the bore. This could be related to flank wear, which also appeared at around 2 min. Thereafter, the growth rate of delamination slowed down. This phenomenon was also observed by other authors [46,61].

Figure 11 shows the influence of the cutting parameters on the life of the drill bit, expressed in cutting time in Figure 11a and quantified in the number of holes drilled in Figure 11b.

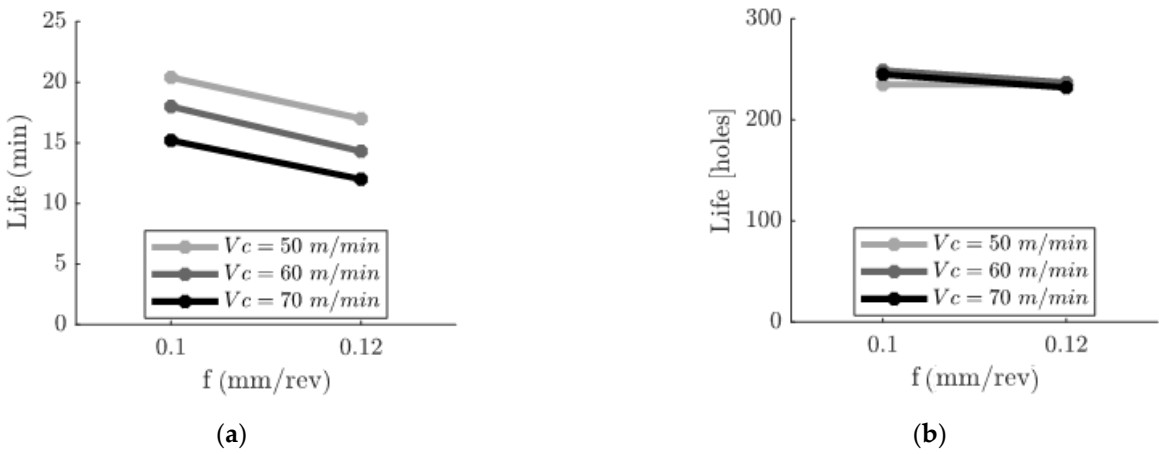

(**a**)                                       (**b**)

**Figure 11.** Tool life based on a critical flank extension of 0.4 mm expressed in (**a**) cutting time and (**b**) holes drilled.

Increasing the feed and the cutting speed had a negative effect on the life of the tool. Specifically, an individual increment of 20% in cutting speed or in feed implied a reduction on the order of 15%. Hence, the cutting conditions that led to the longer tool life, expressed

in cutting time, were the ones with a minimum cutting speed and feed ($V_c = 50$ m/min and $f = 0.10$ mm/rev).

Regarding the tool life expressed in the number of holes drilled, all the cutting conditions tested reached a similar value, with differences smaller than 7%. The cutting speed and feed had a minor effect on the number of holes that could be drilled per cutting edge as explained in Section 3.2. Nevertheless, the parameters of $V_c = 50$ m/min and $f = 0.10$ mm/rev allowed us to perform the maximum number of holes as well, reaching up to 249.

On the other hand, although the tool life expressed in the number of holes drilled was 7% smaller (232 holes) for the parameters $V_c = 70$ m/min and $f = 0.12$ mm/rev, the cutting time was 40% smaller. Thus, considering productivity criteria, it will be interesting to evaluate the use of high values for the cutting parameters. However, a specific economic analysis should be carried out considering quantities such as the cost associated with each tool change and in general, the hourly cost of the production system.

## 4. Conclusions

This research approached the optimization of the one-shot drilling (OSD) strategy of CFRP machining in the assembly of aircraft components through a combined, comprehensive, and in-depth investigation of the tool wear, hole delamination, and evolution of the thrust force and torque. The effect of the cutting parameters on tool life was carried out to select those cutting conditions that had the minimum impact on a real production system, using tools and workpieces from an aeronautical assembly process.

In the tests carried out, the same wear mechanisms and particularities were obtained, very similar to that undergone by the smaller-diameter drills investigated in other studies. Nevertheless, there were some disparities in the evolution of each of them because of the interaction of the fibers with the rake surface. The thrust force and torque analysis revealed that the higher the cutting speed, the greater the growth of the flank extension with cutting time. In the same way, an increase in feed also led to more accelerated wear. In addition, it was also observed that this wear mechanism affected the axial force evolution to a greater extent compared to torque.

The delamination at the hole exit showed no influence of the cutting parameters in the ranges analyzed, so the end-of-life criterion was not established based on the delamination factor, but the extension of the wear suffered by the main cutting edge of the first step.

All the cutting conditions tested reached a similar value of tool life expressed in terms of holes drilled, with differences smaller than 7%, the conditions $V_c = 50$ m/min and $f = 0.10$ mm/rev being those that managed to make the greatest number of holes. However, expressed in time, for the cutting conditions $V_c = 70$ m/min and $f = 0.12$ mm/rev, the same number of holes were completed within 40% of the cutting time.

It is important to keep in mind that the increase in cutting speed and feed moderately reduced the tool life expressed as the number of holes per cutting edge. Therefore, considering productivity criteria, it will be interesting to evaluate the use of high values of cutting parameters. However, a specific economic analysis should be carried out considering quantities such as the cost associated with each tool change and in general, the cost of occupancy of the production system.

Although throughout this research, efforts have been made to emulate the actual drilling conditions of the aircraft assembly phase, when transferring the results to the industrial environment, the variability of the process must be considered. Before performing drilling operations, three main preliminary steps are carried out that might influence the machining performance: a temporary clamping to prevent movement between the parts, thus avoiding assembly defects, and a backlash management to avoid friction during assembly. Following this, the application of a sealant prevents the penetration of moisture from water or other liquids and avoids galvanic corrosion. This imposes additional challenges that are difficult to emulate in the laboratory.

**Author Contributions:** J.F.-P.: conceptualization, investigation, visualization, writing—review and editing; C.D.-M.: writing—original draft, visualization, investigation; M.H.M.: formal analysis, resources, funding acquisition; J.L.C.: writing—review and editing, supervision, project administration. All authors have read and agreed to the published version of the manuscript.

**Funding:** The authors acknowledge the financial support of Airbus Defense and Space through the project Drilling Processes Improvement for Multi Material CFRP-Al-Ti Stacks, to the Ministry of Economy and Competitiveness of Spain through a grant with reference PTA2015-10741-I and project DPI2017-89197-C2-1-R, to the State Investigation Agency through the project Analysis of Defects in Fiber-Reinforced Laminates Due to Manufacturing Processes and Effect on Fatigue Behavior (PID2020-118480RB-C22) and the project Digitalization of Industrial Drilling Process (PDC2021-121368- C21), the Regional Ministry of Education, Youth and Sports of the CAM and the European Social Fund for funding the Aid for the Hiring of a Research Assistant (PEJ-2020-AI/IND-18025) and the MCIN/AEI/10.13039/501100011033 and the European Union "NextGenerationEU"/PRTR" (PDC2021-121368-C21).

**Data Availability Statement:** Not available.

**Conflicts of Interest:** The authors declare no conflict of interest.

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
