# Peer review of "Analysis of Tool Wear and Hole Delamination for Large-Diameter Drilling of CFRP Aircraft Fuselage Components: Identifying Performance Improvement Drivers and Optimization Opportunities"

_jmmp, doi:10.3390/jmmp7020076_

Round 1

Reviewer 1 Report

This paper deals with an important issue for the aeronautical industry and especially the assembly process. It is thus of great interest.

However, the paper must be improved before publication on some points.

1. Concerning the state-of-the art, the bibliography must be improved by studying the work done in the same context (CFRP drilling in the aeronautical field, large-diameter tools, specific tools used in real industrial applications, presence of a glass ply): 

- Drilling of thick composite materials using a step gundrill, Composites part A, vol.103, pp.304-313, 2017.

- Priarone P, Robiglio M, Melentiev R, Settineri L. Diamond drilling of carbon fiber reinforced polymers: influence of tool grit size and process parameters on workpiece delamination. Procedia CIRP 2017;66:181–6.

- Khashaba UA, El-Keran AA. Drilling analysis of thin woven glass-fiber reinforced epoxy composites. J Mater Process Technol 2017;249:415–25.

- Effect of adding a woven glass ply at the exit of the hole of CFRP laminates T on delamination during drilling, Composites part A: Applied Science and Manufacturing, vol.129, 105731, 2020.

2. Also in the field of aeronautical assembly, the tool-life is defined through the respect of all the specifications (diameter, roughness, delamination, burr height for metals…); it is to say that the end-of-tool-life is defined when the last drilled hole doesn’t reach anymore the specification (and then safety coefficients are considered). Thus, it is not directly linked to the tool wear. It must be mentioned.

Below this, as a quality criterion, only the delamination factor is considered in the paper. Other criteria as diameter, cylindricality and roughness must be considered.

Concerning the diameter, as it is measured, its evolution must be presented and discussed.

Also, it is mentioned that "three measurements were taken at the entrance of the hole and three at the exit and the average between them was taken as the final value". From my point of view, this is not acceptable for different reasons. First, this doesn’t correspond to the industrial standard for hole diameter measurement. Usually, 4 points are measured at different levels. Secondly, only the averaged diameter is considered. But measuring 4 points at different levels allows to check the averaged diameter, but also the roundness and the cylindricality of the hole. 

Concerning roughness, it is mentioned that measurements are performed using an internal micrometers. It can be mentioned that, nowadays and generally speaking, no roughness measurements are performed anymore on holes in laminates. It is due to the difficulty to have a representative evaluation of the surface texture. For this, it is interesting to include the following study from Landon Y. and Cherif M.: Characterization of the surface quality of holes drilled in CFRP laminates, Advanced Materials Research, Vol. 698 (2013) pp 107-116. doi:10.4028/www.scientific.net/AMR.698.107. 

Whatever, the analysis of the surface texture remains interesting in this kind of study. But it has to be done considering the 3D texture and associated 3D criteria, without any filtering of the surface. The authors shall consider this comment as an advice for further work.

Finally, concerning delamination, the measurement technique of the damaged diameter must be explained.

3. Some corrections about the analysis of tool-wear must be made: 

First it is mentioned that "The main wear mechanism is located at the flank, combined with small breakages in the coating." This affirmation hides what is the main consequence of tool wear that is to say the edge rounding. This sentence must be modified to be more explicit. 

Also, it would be of importance to measure the evolution of the edge radius during the tests. 

Secondly, it is mentioned that "The higher the cutting speed, the more accelerated the growth of the damage extent." Again this affirmation must be detailed: it is true when considering the evolution of the wear in respect to the cutting time, but it is not true in respect to the removed material volume (or number of holes here). It has to be specified, presented and discussed.

In order to better understand the evolution of the tool wear during cutting, the different states of wear at the different stages must be presented and discussed.

The results presented (Fig.11) show that finally the cutting parameters have no significant impact on tool-life (in terms of "meters cut" or "number of holes" - same thickness).  Does it mean that the study gave no usable result? Maybe a larger range had to be considered for the cutting conditions?

It is concluded that the best choice would be the maximum values for Vc and f as it corresponds to the highest production rate. Again I do not agree with this choice as the criterion is not only the production rate but more generally the cost (as rightly mentioned by the authors in the introduction). And using maximal values for the cutting conditions means needing more tools for the same number of holes and so increasing the global tool cost. So this conclusion must be given as an option but explaining all the consequences.

4. Thrust force and torque must also be plotted in respect to the removed material volume (or number of holes here) and it has to be discussed.

5. A "mechanistic model" is proposed. What is the interest of this model? From my point of view, it doesn’t give any key to understanding about the wear mechanisms and it doesn’t present any interest for industrial application. This section could be deleted. 

6. Finally, the manuscript must be proofread and the English must be improved for some sentences.

Author Response

This paper deals with an important issue for the aeronautical industry and especially the assembly process. It is thus of great interest.

However, the paper must be improved before publication on some points.

1. Concerning the state-of-the art, the bibliography must be improved by studying the work done in the same context (CFRP drilling in the aeronautical field, large-diameter tools, specific tools used in real industrial applications, presence of a glass ply): 

  • Drilling of thick composite materials using a step gundrill, Composites part A, vol.103, pp.304-313, 2017.
  • Priarone P, Robiglio M, Melentiev R, Settineri L. Diamond drilling of carbon fiber reinforced polymers: influence of tool grit size and process parameters on workpiece delamination. Procedia CIRP 2017;66:181–6.
  • Khashaba UA, El-Keran AA. Drilling analysis of thin woven glass-fiber reinforced epoxy composites. J Mater Process Technol 2017;249:415–25.
  • Effect of adding a woven glass ply at the exit of the hole of CFRP laminates T on delamination during drilling, Composites part A: Applied Science and Manufacturing, vol.129, 105731, 2020.

Thank you for your comment, we appreciate it. The suggested studies and other similar ones have been discussed in the Introduction section.

2. Also in the field of aeronautical assembly, the tool-life is defined through the respect of all the specifications (diameter, roughness, delamination, burr height for metals…); it is to say that the end-of-tool-life is defined when the last drilled hole doesn’t reach anymore the specification (and then safety coefficients are considered). Thus, it is not directly linked to the tool wear. It must be mentioned.

Thank you for your comment. This aspect has been mentioned in the Introduction section.

During the assembly of aeronautical components, drilling operations play a critical role, and the lifespan of the tools used in these operations is determined by various factors such as diameter, roughness, and delamination. Typically, the end of a tool's life is defined by the point at which the last drilled hole no longer meets the required specifications. This study specifically focuses on evaluating the quality of drilled holes in relation to delamination.

Below this, as a quality criterion, only the delamination factor is considered in the paper. Other criteria as diameter, cylindricality and roughness must be considered.

Concerning the diameter, as it is measured, its evolution must be presented and discussed.

Also, it is mentioned that "three measurements were taken at the entrance of the hole and three at the exit and the average between them was taken as the final value". From my point of view, this is not acceptable for different reasons. First, this doesn’t correspond to the industrial standard for hole diameter measurement. Usually, 4 points are measured at different levels. Secondly, only the averaged diameter is considered. But measuring 4 points at different levels allows to check the averaged diameter, but also the roundness and the cylindricality of the hole. 

Concerning roughness, it is mentioned that measurements are performed using an internal micrometers. It can be mentioned that, nowadays and generally speaking, no roughness measurements are performed anymore on holes in laminates. It is due to the difficulty to have a representative evaluation of the surface texture. For this, it is interesting to include the following study from Landon Y. and Cherif M.: Characterization of the surface quality of holes drilled in CFRP laminates, Advanced Materials Research, Vol. 698 (2013) pp 107-116. doi:10.4028/www.scientific.net/AMR.698.107. 

Thank you for your comments, we appreciate it. It has been clarified in the document that the scope of the study does not include a complete evaluation of the quality of the hole but only its delamination as it is the main quality problem observed in the industrial process corresponding to the drilling conditions tested in this study. Therefore, the title of the article and the abstract have been modified and this point has been clarified in the Materials and Methods section.

The holes were inspected with a C-Scan and no internal damage was observed. Only delamination and uncut fibers were observed at the exit of the hole, so the delamination factor was used to quantify it.

It should be noted that the hole diameter measurement is only used to determine the delamination factor, but no specific quality control of the hole diameter is performed, so the industry standard of 4 measurements per hole has not been applied. The measurements were performed according to Airbus norms and standards.

Whatever, the analysis of the surface texture remains interesting in this kind of study. But it has to be done considering the 3D texture and associated 3D criteria, without any filtering of the surface. The authors shall consider this comment as an advice for further work.

Thank you for your comment, we appreciate it. It will be taken into account for future publications.

Finally, concerning delamination, the measurement technique of the damaged diameter must be explained.

Thank you for your comment. This aspect has been clarified in Section 2.3 of the document.

3. Some corrections about the analysis of tool-wear must be made: 

First it is mentioned that "The main wear mechanism is located at the flank, combined with small breakages in the coating." This affirmation hides what is the main consequence of tool wear that is to say the edge rounding. This sentence must be modified to be more explicit. 

Also, it would be of importance to measure the evolution of the edge radius during the tests. 

Thank you for your comment. In Section 3.1 of the document, according to Figures 2 and 3, it has been clarified that no rounding of the tool edges has been observed.

Secondly, it is mentioned that "The higher the cutting speed, the more accelerated the growth of the damage extent." Again this affirmation must be detailed: it is true when considering the evolution of the wear in respect to the cutting time, but it is not true in respect to the removed material volume (or number of holes here). It has to be specified, presented and discussed.

Thank you for your comment. The authors agree with your comment. This information has been clarified in the document.

The higher the cutting speed, the greater the extent of damage relative to the cutting time.

In order to better understand the evolution of the tool wear during cutting, the different states of wear at the different stages must be presented and discussed.

Thank you for your comment. This point has been discussed in Section 3.1 of the manuscript.

The evolution of the types of wear described is progressive and homogeneous throughout the test, giving rise to a relatively linear evolution of the degree of flank wear with cutting time, as shown in Figure 4. There is an initial stage of relatively rapid wear called edge settling and an intermediate zone of linear growth of the wear level with some oscillations in the slope due to the fact that only 4 evaluations are performed per tool.

The results presented (Fig.11) show that finally the cutting parameters have no significant impact on tool-life (in terms of "meters cut" or "number of holes" - same thickness).  Does it mean that the study gave no usable result? Maybe a larger range had to be considered for the cutting conditions?

Thank you for your comment. This point has been clarified in Section 3.3 of the document according to the results obtained in the thrust and torque analysis.

It is concluded that the best choice would be the maximum values for Vc and f as it corresponds to the highest production rate. Again I do not agree with this choice as the criterion is not only the production rate but more generally the cost (as rightly mentioned by the authors in the introduction). And using maximal values for the cutting conditions means needing more tools for the same number of holes and so increasing the global tool cost. So this conclusion must be given as an option but explaining all the consequences.

Thank you for your comment. This point has been clarified in the Conclusions.

It is important to keep in mind that the increase in cutting speed and feed rate reduces moderately the tool life expressed in the number of holes per cutting edge. Therefore, considering productivity criteria, it will be interesting to evaluate the use of high values of cutting parameters. However, a specific economic analysis should be carried out considering magnitudes such as the cost associated with each tool change and in general the cost of occupancy of the production system.

4. Thrust force and torque must also be plotted in respect to the removed material volume (or number of holes here) and it has to be discussed.

Thank you for your comment. It has been clarified in the document that the evolution of the thrust force and the torque with cutting time shows that, especially the thrust force, it would be a suitable magnitude to monitor the wear level of the tool in real-time. For this analysis, it is clearer to show the evolution of the thrust force and the torque about the cutting time, just as it is done with the evolution of tool wear.

For this reason, the monitoring of the axial force in these drilling processes could be used to predict the level of tool wears in real time.

In the tests carried out, the evolution of thrust force and torque with the volume of material removed shows similar trends, with changes only in the slopes of the different curves, but without clear trends that could provide relevant information. To achieve greater clarity in the presentation of the main conclusions of this section and once they have been clarified in the document, it has been considered appropriate not to include the graphs of the evolution of the thrust force and torque concerning the volume of material removed.

5. A "mechanistic model" is proposed. What is the interest of this model? From my point of view, it doesn’t give any key to understanding about the wear mechanisms and it doesn’t present any interest for industrial application. This section could be deleted. 

 Thank you for your comment. The sections relating to mechanistic models have been removed from the document.

6. Finally, the manuscript must be proofread and the English must be improved for some sentences.

Thank you for your comment. The document has been revised again and typos, passive voice misuse, wordy sentences and unclear sentences have been corrected.

Reviewer 2 Report

Performance optimization in the drilling of CFRP aircraft fuselage components for special large-diameter drills based on evaluation of tool wear and hole quality

The manuscript deals with the analysis of the drilling process on carbon-reinforced polymer composite panels for aeronautic applications. The topic and the analyses are not extremely innovative, nevertheless, the relevance of the process and its application constitutes a reason for the interest of the readership of the Journal of Manufacturing and Materials Processing.

In the referee’s opinion, the authors perform a careful analysis of the process and of the tool wear to identify the improvement drivers among the process parameters inspected and in the operative window observed, but they do not perform an actual optimization. Therefore, in the referee’s humble opinion, the title and the declaration of the scope of the work (lines 111-113) should be revised.

As a referee, I appreciate the analyses conducted, but I cannot detect the relevance of the model presented in sections 2.4 and 3.4. It does not provide a direct value to the study and, probably due to this reason, it has been scarcely discussed by the authors. Moreover, the parameters without a confidence range for each of them cannot be directly used by other authors or researchers in their future works. My suggestion is to remove this weak aspect from this work.

Please, add a Conclusions section. Some conclusions are reported (wrongly) in the Discussion section.

Author Response

The manuscript deals with the analysis of the drilling process on carbon-reinforced polymer composite panels for aeronautic applications. The topic and the analyses are not extremely innovative, nevertheless, the relevance of the process and its application constitutes a reason for the interest of the readership of the Journal of Manufacturing and Materials Processing.

1. In the referee’s opinion, the authors perform a careful analysis of the process and of the tool wear to identify the improvement drivers among the process parameters inspected and in the operative window observed, but they do not perform an actual optimization. Therefore, in the referee’s humble opinion, the title and the declaration of the scope of the work (lines 111-113) should be revised.

Thank you for your comment, we really appreciate it. It has been clarified in the document the scope of the study. Therefore, the title of the article, the abstract and the lines 111-113 have been modified.

2. As a referee, I appreciate the analyses conducted, but I cannot detect the relevance of the model presented in sections 2.4 and 3.4. It does not provide a direct value to the study and, probably due to this reason, it has been scarcely discussed by the authors. Moreover, the parameters without a confidence range for each of them cannot be directly used by other authors or researchers in their future works. My suggestion is to remove this weak aspect from this work.

Thank you for your comment. The sections relating to mechanistic models have been removed from the document.

3. Please, add a Conclusions section. Some conclusions are reported (wrongly) in the Discussion section.

Thank you for your comment, we really appreciate it. The Conclusions Section has been added, clarifying some of the points presented therein.
